# Moving Object Detection in Freely Moving Camera via Global Motion Compensation and Local Spatial Information Fusion

**DOI:** 10.3390/s24092859

**Published:** 2024-04-30

**Authors:** Zhongyu Chen, Rong Zhao, Xindong Guo, Jianbin Xie, Xie Han

**Affiliations:** 1School of Computer Science and Technology, North University of China, Taiyuan 030051, China; b20220703@st.nuc.edu.cn (Z.C.); b1907073@st.nuc.edu.cn (R.Z.); gxd@sxau.edu.cn (X.G.); kedachang@126.com (J.X.); 2Shanxi Key Laboratory of Machine Vision and Virtual Reality, Taiyuan 030051, China; 3Shanxi Province’s Vision Information Processing and Intelligent Robot Engineering Research Center, Taiyuan 030051, China

**Keywords:** motion object detection, inter-frame transformations, local spatial, optical flow

## Abstract

Motion object detection (MOD) with freely moving cameras is a challenging task in computer vision. To extract moving objects, most studies have focused on the difference in motion features between foreground and background, which works well for dynamic scenes with relatively regular movements and variations. However, abrupt illumination changes and occlusions often occur in real-world scenes, and the camera may also pan, tilt, rotate, and jitter, etc., resulting in local irregular variations and global discontinuities in motion features. Such complex and changing scenes bring great difficulty in detecting moving objects. To solve this problem, this paper proposes a new MOD method that effectively leverages local and global visual information for foreground/background segmentation. Specifically, on the global side, to support a wider range of camera motion, the relative inter-frame transformations are optimized to absolute transformations referenced to intermediate frames in a global form after enriching the inter-frame matching pairs. The global transformation is fine-tuned using the spatial transformer network (STN). On the local side, to address the problem of dynamic background scenes, foreground object detection is optimized by utilizing the pixel differences between the current frame and the local background model, as well as the consistency of local spatial variations. Then, the spatial information is combined using optical flow segmentation methods, enhancing the precision of the object information. The experimental results show that our method achieves a detection accuracy improvement of over 1.5% compared with the state-of-the-art methods on the datasets of CDNET2014, FBMS-59, and CBD. It demonstrates significant effectiveness in challenging scenarios such as shadows, abrupt changes in illumination, camera jitter, occlusion, and moving backgrounds.

## 1. Introduction

With the widespread use of surveillance cameras in recent years, the number of recorded videos has been increasing dramatically. Surveillance videos typically contain complex scene information, and it is critical to reasonably analyze and process the interesting part. In most cases, MOD serves as the foundation for advanced vision tasks, such as video surveillance [1], object tracking [2,3], and human–computer interaction surveillance [4]. It is an important component in intelligent security surveillance video because it provides basic technical support for massive video analysis and processing.

MOD techniques vary depending on whether the camera is moving or not. Numerous methods have been researched for the detection of moving objects when using fixed cameras, including the background subtraction (BGS) methods [5,6,7,8,9], the frame difference methods [10,11], the optical flow methods [12], and deep learning [13,14,15,16,17,18]. They primarily detect motion objects by acquiring inter-frame information or constructing background models. Accurate detection results can be obtained when the background is completely static or the scene is relatively simple. However, in the case of a freely moving camera, the background is typically dynamic and extremely complex. The pixel intensity at a specific location in the background may undergo irregular variations, which may invalidate algorithms designed for fixed cameras, posing a new challenge to MOD.

Currently, there are various MOD methods used to deal with complicated background sceneries, such as camera jitter [19], small object recognition [20,21], abrupt changes in illumination [22], shadow removal [23], and video with noise or a damaged image frame [24]. According to whether the background is modeled or not, it can be broadly divided into two categories, i.e., model-based and motion-based algorithms. The first category combines motion compensation methods [25] to extend MOD under fixed cameras, establishing local or global background models. Model-based algorithms [26,27,28] align several adjacent image frames using affine transformations and then extract foreground objects using background modeling or background subtraction. Since the motion between adjacent image frames is localized, it is challenging to detect slowly moving or temporarily stationary objects. Moreover, this approach fails when the video contains huge amounts of global motion (e.g., panning, tilting, rotating, and jittering). It is due to the fact that the motion vector between consecutive frames does not adequately depict global motion and does not give sufficient information to rebuild the current frame. The second category analyzes foreground objects by estimating the consistency of pixel point or image block motion using the optical flow method [29]. In this case, the MOD results are hardly affected by background variations as the motion-based algorithms [4,30,31] do not require the construction of an explicit background model. These methods perform well when there are relatively large objects and the background is moving in a globally consistent manner on the image. The detection performance, however, is heavily dependent on the accuracy of the motion vector between the foreground object and the background. Moreover, in the presence of local spatial noise and illumination interference, such as changes in the position of light sources, shadows, etc., the algorithm may extract inaccurate motion information, which can affect the subsequent calculation and analysis.

In this paper, we present a hybrid approach that combines the advantages of motion compensation and optical flow for motion object detection. The primary focus is on enhancing the accuracy of global motion compensation and mitigating the interference of complex backgrounds on local foreground areas. Therefore, we propose a MOD algorithm based on global motion compensation and local spatial information fusion on a freely moving camera. On the premise of eliminating inter-frame motion transformation with large errors, we convert the relative projection transformation problem into a global-based absolute projection transformation problem and perform global optimization to ensure accurate transformation. Furthermore, in the local spatial, the optical flow segmentation approach is combined to enhance the foreground object contour and reliably detect the foreground object. The contributions are summarized as follows:(1)A novel motion compensation framework for global optimization is constructed. This framework aims to estimate inter-frame transformations effectively and improve global matching accuracy by optimizing the residual matching and drawing on the use of back-end optimization methods [32] and STN [33].(2)A novel method for local spatial information fusion is proposed, which utilizes several types of local spatial boundary information, to effectively handle challenges such as foreground false alarms, hollow foreground objects, and unclear contours.(3)An end-to-end MOD method is proposed through the collaborative work of the two modules mentioned above. The proposed approach is validated on three datasets, and a comparison is made with existing methods. The results show state-of-the-art performance and confirm the efficacy of the proposed method. Furthermore, the method covers a wide variety of camera motions, which increases its practical utility.

## 2. Related Works

The process of extracting and identifying changing areas from the background image of a video image sequence is known as MOD. Object detection methods for dynamic scenes are critical as cameras are increasingly used on mobile cameras. Existing methods for detecting motion objects with moving cameras are classified into three types: model-based algorithms, motion-based algorithms, and hybrid algorithms.

### 2.1. Model-Based Algorithms

The model-based algorithm focuses on finding invariant areas to construct a statistical model of the scene by means of motion compensation. Liu et al. [26] proposed an unsupervised learning framework based on the foreground model that computes motion using homography transformations across consecutive frames to find a limited number of keyframes. The motion cues obtained from the Markov Random Field (MRF) are combined with the appearance of the Gaussian Mixture Model (GMM) to achieve motion object segmentation. Similarly, Zamalieva et al. [27] proposed a method to adaptively change the geometric transformation matrix to estimate the background motion caused by camera motion. They utilized Bayesian learning to model the appearance of the background and foreground. Recently, some studies [34] proposed a Poisson-fusion-based background orientation area reconstruction approach to obtain foreground objects by computing the difference between the original orientation area and the reconstructed orientation area. In other methods, low-rank matrix factorization is employed for subspace learning. Eltantawy et al. [20] proposed a new principal component pursuit (PCP) [35] method for modeling moving objects as moving sparse objects based on multiple local subspaces. Others [36] used the developed candidate background model to iteratively update the background model, and then performed background subtraction based on the consistency of local changes. Some methods also focus on establishing background models in the subspace. Chelly et al. [37] used Robust Principal Component Analysis (RPCA) [38] to generate a local background map for background reduction. However, this method is more sensitive to camera rotation and image perspective translation, not to mention that it is not an end-to-end learning approach, and its applicability is limited. By embedding affine transformation operators in online subspace learning, He et al. [19] obtained a more extensive subspace alignment. However, the above algorithm cannot handle non-rigid dynamic backgrounds, and its performance may suffer if complicated backgrounds cannot be approximated as planes.

### 2.2. Motion-Based Algorithms

Motion-based algorithms use the difference in motion patterns between foreground and background objects. It is classified into two types: point-based and layer-based. Point-based methods detect and track sparse feature points before executing segmentation on the same type of object. Some studies [39] deconstruct the motion trajectory matrix and classify foreground objects at the pixel level using trajectory information. Some [40,41] extract foreground areas from the video by clustering the obtained long feature trajectories. Manda et al. [42] combined optical flow information between neighboring frames to extract features of the object in both spatial and temporal dimensions. Others utilized [43] lengthy trajectory features and a Bayesian filtering framework to estimate motion and appearance models. Although these methods are robust in handling large scenes of camera motion, they only create sparse point segmentation that must be post-processed to produce dense segmentation.

The layer-based method computes dense optical flows and then clusters them based on motion consistency. Shen et al. [30] used dense optical flow to calculate the trajectory of each pixel point and then used a two-stage bottom–up clustering approach to identify the final motion object. Some researchers utilize initial motion boundary information generated by dense optical flow fields to detect moving objects. Sugimura et al. [31] employed the initial foreground and background label boundary information in conjunction with the area segmentation method to recognize foreground objects. The initial labels are computed using the motion boundaries of two different flow field sizes and directions. Additionally, Zhang et al. [44] used the weighted local difference in orientation of the optical flow to detect the moving object’s rough contours, and it is used to guide the selection of prospect objects. In contrast, Bideau et al. [4] combined the optical flow angle and size to optimize the difference between object motions. Due to quick camera movements and video noise, such methods may create some mistakes in measurements. Singh et al. [45] used the similarity of image and optical flow feature information to generate edge weights, and then self-supervised trained the network using a graph cut mask. The optical-flow-based approach is further constrained by the initialization process, which necessitates the development of a priori information such as contour lines and object numbers. Recently, deep learning methods [46,47,48,49,50,51,52] have been used to extract optical flow features to detect moving objects.

### 2.3. Hybrid Algorithms

With the intention of improving detection, hybrid approaches integrate information such as motion, color, and appearance. In this regard, Elqursh et al. [43] relied on long trajectory information in low-dimensional space, considering non-spatial trajectories as foreground objects. They utilized long trajectories, motion, and appearance models combined within a Bayesian filtering framework to obtain the final foreground objects. Delibacsouglu et al. [53] proposed a moving object detection method based on background modeling and subtraction, which represents the background as a model with features such as color and texture. Similar to the above method, Cui et al. [54] used Markov stochastic models for trajectory, appearance, and spatiotemporal cues for the detection of moving object trajectories and background trajectories generated by optical flow. Reference [55] created foreground probability estimates by fusing the motion and appearance modules and then used graph cuts to generate the final segmentation masks. Some researchers also proposed to maximize the relationship between foreground and background clues as much as possible. Makino et al. [56] combined the fraction obtained from the difference between the background model and the current frame with the motion fraction obtained from the local optical flow calculation to more accurately detect moving objects. This method, however, is ineffective when the moving object is too large. Particularly, some studies [57] modeled multiple foreground objects appearing in the scene at different levels, treating it as a semantic segmentation problem. They estimated the motion and appearance models of these objects and employed a Bayesian filtering framework to infer a probability map. In another research work, Zhao et al. [58] used the GMM to obtain the confidence of the foreground cue for each pixel point after motion compensation. The confidence in the foreground cue combines with the confidence in the background pixel points obtained by feature point matching. Finally, they integrated the two confidences to achieve an accurate segmentation result.

## 3. Method

In this section, we propose a MOD method for a moving camera. This method aims to adapt to complex environments for detecting foreground objects and enhance the overall detection performance. Traditional foreground object detection methods are often susceptible to environmental factors, such as camera motion and illumination transformations. These factors can lead to inaccurate detection results. To solve this problem, we use a novel motion compensation global optimization framework to reduce the effect of camera motion on detection results. Additionally, we address the influence of local spatial lighting variations by constructing a local spatial background model and employing a spatial information fusion strategy. An overview of the algorithm is shown in Figure 1.

### 3.1. Inter-Frame Registration

The projection transformation establishes the correspondence between the points $\left(x,y\right)$ and $\left(\hat{x},\hat{y}\right)$ in two different image frames, which is shown below:(1)$$\tau \hat{V}=H^{T}V$$
where $\hat{V}={\left[\hat{x},\hat{y},1\right]}^{T}$ and $V={\left[x,y,1\right]}^{T}$ represent the vector form of the point, and τ is a random scaling constant. The projection transformation matrix is denoted as $H\in R^{3\times 3}$. Given η>3 and ${\left\{V_{i}\to {\hat{V}}_{i}\right\}}_{i=1}^{\eta }$, the least squares method is used to estimate the optimal transformation to *H*, which is shown below:(2)$$\min_{h}{∥\psi h∥}^{2}\;s.t.\;h_{9}=1$$
where *h* is the vectorize representation of *H*, with the constraint that the last element $h_{9}$ is equal to 1. $\psi^{T}=\left[\psi_{1}^{T},\ldots ,\psi_{\eta }^{T}\right]$. The solution of Equation (Equation 2) is the smallest right singular vector of ψ, scaled so that the last element is 1, and the following equation holds:(3)$$\psi_{i}=\left[\begin{array}{ccc} 0 & V_{i}^{T} & -{\hat{y}}_{i}V_{i}^{T}\\ V_{i}^{T} & 0 & -{\hat{x}}_{i}V_{i}^{T} \end{array}\right]\in R^{2\times 9}$$

During the registration process, SURF [59] is used to compute features in each frame of the image, and then RANSAC [60] is employed to eliminate point matches with significant errors, aiming to estimate *H* with minimal loss.

In order to achieve precise inter-frame matching pairs while generating more overlapping areas, it is important to ensure that the local RPCA segments are more comprehensive foreground objects. Inter-frame matching extends to each image with its next *n* frames, as shown in Figure 2. The projection transformation error ϕ is calculated for each frame with its next *n* frames, and the matching pair $N_{\max}$ with the largest error value will be rejected, as shown in Equation (Equation 4):(4)$$N_{\max}=\arg\max_{n}\frac{M_{I\to I_{n}}}{C_{I\to I_{n}}}\;$$
where $\phi =M_{I\to I_{n}}/C_{I\to I_{n}}$ and $\left\{I,I+1,\ldots ,I+n\right\}$ are the current frame and the following *n*th frame images. $I_{n}$ represents the image obtained by the projective transformation of the current frame *I* with respect to the frame I+n. $C_{I\to I_{n}}$ denotes the number of overlapping pixels of $I_{n}$ and I+n. $M_{I-I_{n}}$ denotes the number of pixels in the overlapping pixels that satisfy the threshold of the difference between the two pixels greater than ρ.

### 3.2. Global Projection Optimization

The motion compensation for inter-frame motion using the transformation matrix *H* generated by the preceding method frequently has substantial inaccuracies. To improve the transformation accuracy even further, the registration images are transferred to the coordinate system using the intermediate frame as a reference, and the registration data are processed together. It will solve the synchronization problem on the special Euclidean group by optimizing the mentioned inter-frame relative transformation matrix with a back-end optimization algorithm. Given a set of noisy paired transformation matrices, the values of an unknown pose (position and orientation in Euclidean space) are estimated through optimization, as shown in Equation (Equation 5). By utilizing such optimization method, the noisy inter-frame relative transformation matrix ${\tilde{T}}_{ij}$ is globally optimized to obtain the absolute transformation matrix $T_{j}$ with the reference frame being the intermediate frame.
(5)$$\begin{array}{c} {\left(T_{j}\right)}_{j}^{N}=arg\;min\sum_{\left(i,j\right)\in \varepsilon }a_{ij}{∥R_{i}-R_{j}{\tilde{R}}_{ij}∥}_{F}^{2}+b_{ij}{∥t_{i}-t_{j}-R_{j}{\tilde{t}}_{ij}∥}_{l2}^{2} \end{array}$$
Here, the inter-frame information is denoted as a non-linear weighted undirected graph G=(ν,ε,ω), and ν denotes the finite set of optimized frames, $\left(i,j\right)\in \varepsilon$ denotes the matching relationship between the *i*-th frame and the *j*-th frame, and the weights *w* store information about the inter-frame relative transformation matrix ${\tilde{T}}_{ij}$. *N* is the *n*-th frame of the image. ${∥*∥}_{F}^{2}$ and ${∥*∥}_{l2}^{2}$ are Frobenius norm and L2 norm, respectively. $a_{ij}$ and $b_{ij}$ are optimized weights, while ${\tilde{t}}_{ij}$ and ${\tilde{R}}_{ij}$ (rotation matrix) belong to the planar rotation group $SO\left(2\right)$. ${\tilde{T}}_{ij}=\left[\begin{array}{cc} {\tilde{R}}_{ij} & {\tilde{t}}_{ij}\\ 0_{1\times 2} & 1 \end{array}\right]$ and $T_{j}=\left[\begin{array}{cc} R_{j} & t_{j}\\ 0_{1\times 2} & 1 \end{array}\right]$.

The STN is utilized to optimize the residual transform and minimize the loss in Equation (Equation 6). It has the ability to automatically perform spatial mapping transformations and learn translations, scaling, rotations, and more general transformations. Its input consists of a series of transformation parameters $T_{j}$ and images, and its output is the transformed image. For more information about the benefits of using STN to learn projection transformations, please refer to [61].
(6)$$\Gamma =arg\;min\sum_{j=1}^{N}\sum_{i=1}^{D}\frac{\rho_{i}^{j}f\left(x_{i}^{j}-u_{i},\varphi \right)}{\sum_{a\in D}\rho_{a}^{j}}+\lambda \sum_{j=1}^{N}d\left(T_{j},\theta \right)$$
(7)$$u_{i}=\frac{\sum_{j=1}^{N}\rho_{i}^{j}x_{i}^{j}}{\sum_{j=1}^{N}\rho_{i}^{j}}$$
Here, *N* is the *n*-th frame image, and *D* denotes the panorama resulting from absolutely transforming all image frames after back-end optimization. The value of $\rho_{i}^{j}$ indicates the positional relationship of an image within a panoramic image. It takes a value of 1 when there is overlap between the image transformed by the absolute transformation of the *j*-th frame and the panoramic image, and 0 otherwise. $f\left(.,\varphi \right)$ is a differentiable robust error function. $x_{i}^{j}$ is the pixel value at position *i* of the *j*-th frame of the transformed image, and $u_{i}$ is the average pixel value of all transformed images at point *i* of the panorama. λ>0; the parameter $d\left(*\right)$ is used to control a regularization penalty term, which projects the absolute transformation matrix $T_{j}$ computed from STN to a regular matrix θ.

Figure 3 shows the panoramic background image obtained from the above global motion compensation process. By comparison, it is evident that the proposed method achieves more refined inter-frame motion compensation results and a more complete panoramic background image.

### 3.3. Local Spatial Detection

The foreground–background detection problem in the context of a moving camera can be simplified to the standard static camera foreground–background detection problem through motion compensation. The RPCA is commonly used for foreground object detection. Its core idea is to decompose the noise-containing matrix (video sequence) into a low-rank matrix (background of the video sequence) and a sparse matrix (foreground of the video sequence). Thus, the projected transformed image data can be decomposed as follows:(8)$$\min_{L,S}{∥L∥}_{*}+\zeta {∥S∥}_{1}\;s.t.\;M=L+S$$
where ${∥.∥}_{*}$ represents the nuclear norm of the matrix, ${∥.∥}_{1}$ represents the L1 norm of the matrix, and ζ is the parameter that controls sparsity. The matrix solution process can be optimized using the Alternating Direction Method of Multipliers (ADMM) [62].

When the RPCA is used directly for a panoramic image, it frequently results in excessive computational complexity and insufficient memory. Therefore, the method divides the panorama size into several smaller sliding windows. Each window is shifted by *z* pixels horizontally or vertically, as shown in Figure 4.

All images are transformed by absolute projection to their corresponding positions on the panoramic image. The RPCA is applied to compute the local window background image for the area where the image coverage of the sliding window exceeds κ. During background subtraction, the local windows of the transformed image area are stitched together to obtain a local background image. The subtraction is performed between the transformed image area and the local background image within a range of *k* pixels around it. The minimum difference is computed as the final position, as shown in Equation (Equation 9):(9)$$arg\;min\sum_{i=1}^{D^{\prime }}\left|x_{i-c}-p_{i}\right|$$
where $D^{\prime }$ represents the area obtained after performing absolute projection of the image, *i* denotes the pixel position at a certain point within the area, and *c* represents the offset distance from the pixel point *i*. The valid range for *c* is [0,k]. *x* is the pixel value of the pixel point of the image at position i−c, $p_{i}$ is the local background pixel value corresponding to the *x* position. By varying the value of *c*, the minimum value of Equation (Equation 9) is obtained as the precise adjustment position.

Due to the lack of illumination in the shadow area of moving objects, the pixel values exhibit local inconsistency. We define $T_{h}$ as the consistency of changes between the current frame and the local background, which is used to eliminate shadow areas during the background subtraction process.
(10)$$T_{h}=\frac{1}{N+M}\sum_{i=x-N}^{i=x+N}\sum_{i=y-M}^{i=y+M}{\left(I\left(i,j\right)-B\left(i,j\right)-\delta \left(x,y\right)\right)}^{2}$$
Here, $I\left(i,j\right)$ represents the pixel values at the position of the moving object in the panoramic image, while $B\left(i,j\right)$ represents the pixel values of the local background that are consistent with the position of the moving object. The local difference between two frames is computed by averaging the pixel differences within a rectangular area (N×M) around the corresponding pixels $\left(x,y\right)$. This average is denoted as $\delta \left(x,y\right)$. If $T_{h}$ exceeds a certain threshold, it is considered that the point on the motion object corresponds to a shadow.

### 3.4. Spatial Information Fusion

During local spatial subtraction, there is a possibility of low recall due to the similarity between the object pixel color and the background. By incorporating optical flow segmentation techniques, it is possible to accurately extract pixels of moving objects with high recall, even when their intensities are close to the background. The proposed method effectively integrates optical flow and background spatial information to optimize situations involving foreground false alarms, detection omissions, and unclear boundaries. The proposed method combines optical flow and background spatial information to optimize foreground false alarms, detection misses, and unclear boundaries. It aims to achieve a better balance between sensitivity and specificity in object detection.

Edge segmentation is performed on the coarse foreground and visualized optical flow field maps resulting from the above process. The degree of segmentation is determined based on the changes in the areas (adjacent areas of edge pixels) in the binary image, and the number of areas increases as the segmentation degree deepens. Then, at a different degree of segmentation, the intersection over union (IoU) between the two image areas is calculated based on whether they are adjacent in position. The final segmentation degree is determined by selecting the segmentation degree with the highest IoU. In other words, the binary segmentation degree that achieves the most stable changes in area blocks is selected for refining the foreground with greater precision. Thus, the area block division problem is solved by using the method of element grouping, and each area block is enclosed within a minimum enclosing circle. The binary processing and area segmentation results are shown in Figure 5.

After selecting the binary areas of the visualized optical flow field and the coarse foreground image, the corresponding areas between the two images are evaluated. This evaluation is based on the distance between their centroids and the radius relationship. The areas that satisfy Equation (Equation 11) are retained, while the areas that do not satisfy it are discarded.
(11)$$\alpha \left|d\right|+\beta \left|r_{1}-r_{2}\right|<\sigma$$
Here, *d* is the centroid distance between the corresponding areas of the two images, *r* is the respective radius, and α and β are the weights of the corresponding centroid distance and radius. Since the focus here is on the accuracy of the position, a higher weight is assigned to the centroid distance *d* compared with the weight assigned to the radius. The threshold value is denoted as σ.

Once the correct areas are identified, an iterative segmentation is performed using interactive GraphCut algorithms, such as GraphCut [63], OneCut [64], and GrabCut [65], on the minimum bounding rectangles of the selected areas. This process results in a refined foreground image, as shown in Figure 6.

## 4. Experiments

In this section, we conducted extensive experimental evaluations on three datasets featuring fixed, jittery, and moving cameras, i.e., CDNET2014 [66], FBMS-59 [40], and CBD [67]. These datasets consist of various video sequences capturing both rigid and non-rigid moving objects from PTZ cameras, handheld cameras, and unmanned aerial vehicles (UAVs). Our method was also compared with state-of-the-art methods from recent years through qualitative and quantitative evaluations. Typical images from the experimental dataset are shown in Figure 7. Our experiments are run on a Windows 10 PC with AMD Ryzen 7 4800U with Radeon Graphics 1.8 GHz. The processing time for our computation is approximately several seconds per frame.

### 4.1. Dataset and Metrics

The CDNET2014 [66] dataset is a dedicated video dataset for evaluating MOD methods with relatively fixed video capture devices. The dataset consists of 53 video sequences, categorized into 11 different complex background scenarios encountered in indoor and outdoor surveillance video environments, including Bad Weather (BW), Baseline (BL), Camera Jitter (CJ), Dynamic Background (DB), Intermittent Object Motion (IOM), Low Framerate (LF), Night Videos (NV), Pan–Tilt–Zoom (PTZ), Shadow (SH), Thermal (TH), and Turbulence (TU).

Then, the FBMS-59 [40] dataset is a collection of 59 video sequences specifically designed for motion segmentation. During the video capture process, the camera is in motion, involving translation, rotation, and scaling transformations. Similar to other methods that use partial data from the FBMS-59 dataset as a testing benchmark, the experiments were conducted only on the most commonly used video sequences (cars1-cars8, dogs01, dogs02, people1, and people2) [34,54,57]. There are two main reasons for this: (1) In some video sequences of the dataset, there are multiple moving objects present. However, only a subset of these objects is used for evaluation purposes, as including all of them would introduce additional false positives that could impact the final detection results. (2) A significant portion of the image (>70%) is occupied by foreground objects, which poses initialization challenges for most algorithms.

Then, the CBD [67] dataset is a collection of video sequences designed for MOD. It consists of five different scenes, including crowds, traffic, and other real-world moving objects. The presence of occlusions in the scenes poses significant challenges for MOD.

Similar to other papers, the R (Recall), P (Precision), and Fm (F-measure) metrics have been used for evaluation. R and P are measures of completeness and accurateness, respectively. Fm is a combination of R and P. These metrics are defined as follows:$$R=\frac{TP}{TP+FN}\;P=\frac{TP}{TP+FP}\;Fm=2\times \frac{PR}{P+R}$$
where TP, FP, and FN indicate true positive, false positive, and false negative, respectively. TP is the number of pixels correctly detected as foreground (moving object area). FP is the number of background pixels incorrectly detected as foreground. FN is the number of foreground pixels incorrectly detected as background.

In addition, we set the inter-frame matching number *n* to 5, the overlapping pixel threshold ρ to 15, the penalty term coefficient λ to 0.5, the sparsity control parameter ζ to 0.6, the offset range *k* to 5, α and β to 0.4 and 0.6, and sigma to 10. All parameter settings were fixed for all experiments to ensure the consistency and reproducibility of our results.

### 4.2. Experimental Analysis of MOD under Fixed and Jittered Cameras

#### 4.2.1. Quantitative Analysis

In this paper, our method is compared with state-of-the-art methods, including incPCP [7], SWCD [9], AdMH [10], FBS-ABL [6], t-OMoGMF [5], FgSegNet v2 [18], and BSUV-Net [17] for quantitative analysis under fixed cameras on the CDNET2014 dataset. These methods represent typical, recent mainstream, and deep learning methods. The Fm values of each method in different categories of the CDNET2014 dataset are presented in Table 1. According to Table 1, it can be observed that our method achieves better performance in most video categories. The average performance of our method outperforms all comparison methods, with an Fm of 0.8027. It demonstrates overall robustness, performing well in categories such as BW, BL, DB, PTZ, and SH. Compared with the second-best method BSUV-Net, which performs remarkably well in fixed scenes, our method has increased the average Fm by 0.0159. In the BL category, the BSUV-net achieves an impressive Fm of 0.9693. However, its performance in the PTZ category is less satisfactory, with an Fm of only 0.6282. The decrease in detection accuracy in the PTZ category is due to the non-rigid transformations that foreground objects may undergo in such scenarios. Moreover, the BSUV-Net heavily relies on the quality of semantic information, which needs to be incorporated as prior knowledge during training. Therefore, its detection performance is highly dependent on the quality of semantic information. Meanwhile, the traditional background model-based methods such as SWCD, FBS-ABL, incPCP, AdMH, and t-OMoGMF also exhibit unsatisfactory detection performance in the PTZ category. In the category BW, it can be seen that our method can still accurately detect moving objects under bad weather conditions, with a detection accuracy reaching 0.8873. The proposed method in this paper fully utilizes spatial information for complementary advantages and reduces the impact of non-rigid transformations. The deep-learning-based method FgSegNet v2 shows low overall detection accuracy in unsupervised scenarios, making it difficult to detect well-defined foreground objects.

From Table 1, it can be observed that most methods perform poorly in the NV and IOM categories. This is because the NV category contains strong glare halos due to low visibility. In the IOM category, methods may mistakenly identify long-term stationary foreground objects as background, which significantly affects the detection performance.

Additionally, we specifically analyze the impact of the camera jitter and perform performance analysis using four camera jitter video sequences from the CDNET2014 dataset: badminton, boulevard, sidewalk, and traffic. The quantitative comparative analysis results are shown in Figure 8. According to Figure 8, our proposed method demonstrates favorable performance on the jitter video sequences. Compared with other methods, our approach achieves Fm values above 0.7 for each video sequence class, with an average value of 0.7962, showing a better detection performance. In contrast, the values obtained from AdMH indicate the instability of its detection performance. When there is significant camera jitter and image frame blur in a video, as seen in the “sidewalk” example, the detection performance of most algorithms suffers. In contrast, our proposed method maintains a relatively stable detection level overall.

#### 4.2.2. Qualitative Analysis

Figure 9 shows the foreground results obtained by our method on each category of the CDNET2014 dataset. Our method shows superior performance in detecting foreground object contours for the video categories BL, PTZ, BW, SH, CJ, and DB, independent of external environments such as bad weather. Notably, it exhibits the ability to accurately detect smaller foreground objects, such as LF. In the categories of IOM, TH, and TU, a few instances of false positives were detected. However, overall, our method demonstrates improved robustness in background restoration, as well as reduced noise and shadow effects in video frames. It exhibits strong adaptability to various conditions. The foreground objects detected in the PTZ and NV categories are slightly larger than the ground truth. This is due to the inherent perspective changes that occur as the objects move, causing shape deformations. Additionally, the pixel similarity between the shadow areas of these objects and the background limits the effectiveness of local inconsistency detection, as discrepancies of less than 5 pixels are insufficient to trigger such detections.

The experiment also included testing on the office video sequences from CDNET2014 under different illumination conditions, and the results are shown in Figure 10. It can be observed that the larger the illumination scaling factor |Alpha| value, the clearer the contours shown in the residual images. This indicates a greater disparity in illumination conditions between the illumination images and the adaptive background image. By adaptively processing multiple image frames, our method gradually obtains the most suitable background condition, demonstrating a certain level of resistance to variations in lighting.

### 4.3. Experimental Analysis of MOD under Freely Moving Cameras

Table 2 and Table 3 show a performance comparison of our method under moving cameras with MLBS [57], IFB [58], JA-POLS [37], Sugimura [31], CAG-DDE [51], DS-Net [48], LOCATE [45], and LTA [52] on the FBMS-59 and CBD datasets. CAG-DDE and DS-Net are deep learning methods that use optical flow features as inputs. The average value of multiple measurements is taken as the final Fm. In the IFB algorithm, specifically, the average of the SIFT, SURF, and KAZE feature matching algorithms is used as the final result. Figure 11 qualitatively illustrates the comparative results of various methods on typical frames from the FBMS-59 dataset.

As can be seen from Table 2, our method outperforms the compared methods in most of the video sequences, achieving an overall detection accuracy 0.0179 higher than that of the second-best method. Particularly, it demonstrates strong robustness in the case of rigid motion video sequences, such as cars1–cars8. For non-rigid motion (e.g., people and dogs), the IFB and JA-POLS methods may produce partial false negatives in the foreground due to static areas within the moving objects. Consequently, the Fm values for these methods are only around 0.6. MLBS and Sugimura use hybrid methods to handle moving objects, no longer relying solely on a single technique for detection, which allows them to adapt better to complex environments. Even so, these two methods are not as effective as our method in detecting both rigid and non-rigid motions. Our method effectively combines the advantages of background and optical flow segmentation, mitigating the impact of non-rigid motion to a certain extent. Due to the limited number of video frames, the IFB and JA-POLS methods lack sufficient images to generate statistical models, which affects the extraction of foreground objects. JA-POLS is also prone to the influence of the surrounding environment and background color, leading to the occurrence of holes in the foreground objects. Further, LOCATE relies on optical flow information for detection, thus the obtained object information is coarse. On the other hand, the method LTS primarily relies on pixel changes to detect foreground objects. Therefore, its detection performance may be affected in dynamic background situations, resulting in an overall accuracy of only 0.7067. From Figure 11, it is evident that the proposed method combines the advantages of low-rank matrix decomposition and optical flow to effectively eliminate foreground false alarms and obtain accurate foreground contours. Particularly, for rigid motion (such as cars), it exhibits a better ability to restore its original shape. The methods CAG-DDE and DS-Net, similar in nature, achieve an Fm of over 0.9 for larger objects. However, the detection performance significantly deteriorates when multiple moving objects are present in the scene or when the size of the moving objects is small (such as cars3 and cars5). Furthermore, the performance was evaluated on the occlusion dataset CBD, as shown in Table 3. The proposed algorithm maintains steady performance even in complex background circumstances on mobile cameras. It faces challenges in detecting moving objects with significant occlusions, which proves to be difficult for methods like JA-POLS, CAG-DDE, DS-Net, LOCATE, and LTS.

### 4.4. Analysis of Spatial Fusion Threshold and Ablation Study

#### 4.4.1. Analysis of Threshold σ for Spatial Information Fusion

The spatial relationship in spatial information fusion is determined by the threshold parameter σ in Equation (Equation 10). Figure 12 shows the variation in precision and recall for different σ. From the graph, it can be observed that the accuracy and recall remain relatively stable in the range of 2–18 for σ. This allows Fm to maintain a high level of measurement accuracy. However, once the threshold value exceeds 18, the accuracy starts to decrease significantly. This is because increasing the threshold expands the spatial matching range, causing more background pixels to be classified as foreground objects, leading to an increase in false positives.

#### 4.4.2. Ablation Study

We set up Test1–Test5 to verify the effectiveness of the global motion compensation optimization and the interactive graph cut algorithms (GraphCut, OneCut, and GrabCut) in our method. All three algorithms possess good interactivity and controllability, enabling easy integration with our algorithm to guide them in image segmentation. They all adopt a global optimization strategy, allowing for the consideration of relationships between pixels at a global scale, thereby obtaining accurate segmentation results. We conducted experiments using 12 representative video sequences (cars1–cars8, dogs01, dogs02, people1, and people2) from the FBMS-59 dataset. The qualitative results of the experimental part are illustrated in Figure 13. The quantitative evaluation of Fm values is presented in Figure 14. In addition, the experiments also evaluated the average time performance of the three graph cut methods we used and other graph cut algorithms for single segmentation in the video sequence “cars1” with a resolution of 640 × 480. The results are presented in Table 4.

From Figure 13, it can be observed that, compared with using only projective transformation, our proposed motion compensation optimization method improves the accuracy of foreground object detection and reduces some false positives. Furthermore, the combination with graph cut clearly demonstrates a significant improvement in detection performance. Moreover, as shown in the quantitative results of the algorithm modules in Figure 14, it can be observed that, in terms of segmentation accuracy, the OneCut and GrabCut segmentation algorithms outperform the GraphCut algorithm overall. The performances of OneCut and GrabCut segmentation are comparable, with both achieving similar segmentation results. Table 4 reveals that, under the same experimental conditions, the OneCut algorithm exhibits slightly faster time performance compared with GrabCut. However, considering that OneCut and GraphCut perform single segmentation, while GrabCut allows for multiple iterative segmentations, GrabCut demonstrates better control over details and can handle more complex segmentation scenarios. Therefore, when conducting experiments, a reasonable choice should be made based on the specific circumstances. Moreover, compared with other methods using graph cuts, our approach maintains a considerable detection speed even with limited computational resources (Windows 10 PC with AMD Ryzen 7 4800U with Radeon Graphics 1.8 GHz). Optimization measures such as object pooling and caching were implemented in the code to improve memory usage efficiency (peak memory usage is 341.2 MiB, and incremental memory usage is 227.9 MiB). We believe that these results highlight the robustness and applicability of our approach, even under challenging computational conditions. In the future, running our experiments on a more powerful server will significantly reduce processing times and improve overall performance.

## 5. Conclusions

In this paper, we have proposed a novel motion object detection algorithm for mobile cameras that effectively integrates local and global visual information for foreground/background segmentation. The superiority of the method is demonstrated qualitatively and quantitatively through experiments using three public datasets. The experimental results show that the model performs well in various challenging scenarios, including poor illumination conditions, abrupt changes in illumination, camera jitter, shadows, and both fixed and mobile cameras. The model exhibits good performance and scalability. As our method relies on the differentiation of pixel points to identify continuously moving objects, it is challenging to detect disguised or intermittent moving objects effectively. In future work, we plan to incorporate object feature information such as shape, texture, and motion patterns to complement pixel-level analysis. By leveraging richer feature representations, the more accurate and robust detection of moving objects could be achieved, particularly in scenarios where traditional pixel-based methods may struggle. These enhancements may not only improve the reliability of moving object detection but also extend the applicability of our method to a wider range of dynamic scenes and environmental conditions.

## Figures and Tables

**Figure 1 sensors-24-02859-f001:**
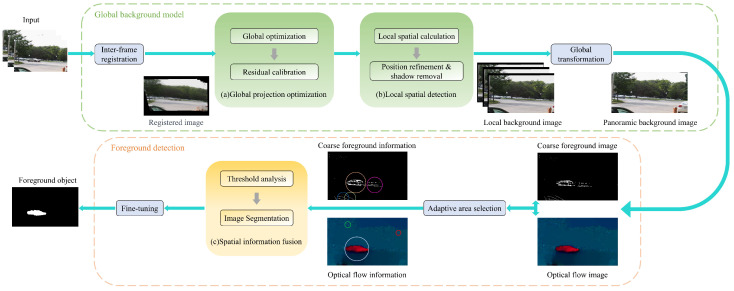
Overview of the proposed method: (1) Input: input the video sequences. (2) The transformation between adjacent frames is optimized globally. Next, to acquire the global background model, we apply the global transform to the local background model obtained from local spatial detection. (3) Combine the coarse foreground image obtained from the global background model with the optical flow information. Then, select the corresponding object area using an adaptive method, and fuse the spatial information from both sources to obtain the final foreground object.

**Figure 2 sensors-24-02859-f002:**
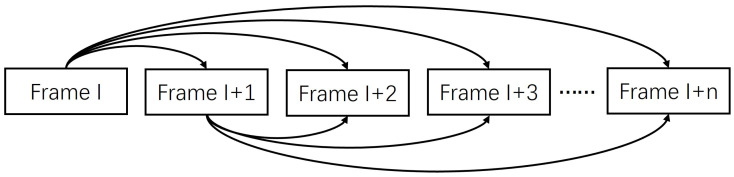
The image registration process.

**Figure 3 sensors-24-02859-f003:**
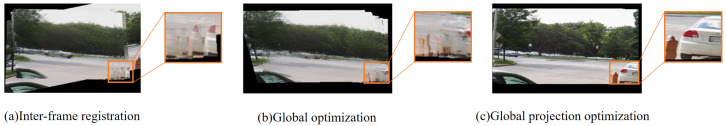
Panoramic background image: (**a**) shows the panoramic image obtained by inter-frame alignment only, (**b**) shows the panoramic image after global optimization, and (**c**) shows the panoramic image after global optimization and residual transformation.

**Figure 4 sensors-24-02859-f004:**
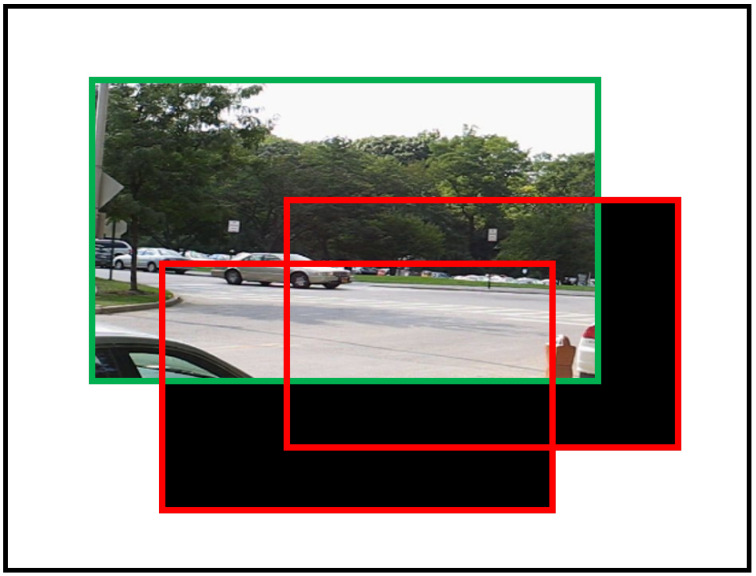
The green bounding box indicates the position of the image frame in the panorama. The red bounding box represents a sliding window, which only computes the overlap between the sliding window and the transformed image (with an overlap area larger than κ). The non-overlapping areas (black areas) within the red bounding box and the areas outside the red bounding box are excluded from the computation. The results are then saved for each window.

**Figure 5 sensors-24-02859-f005:**
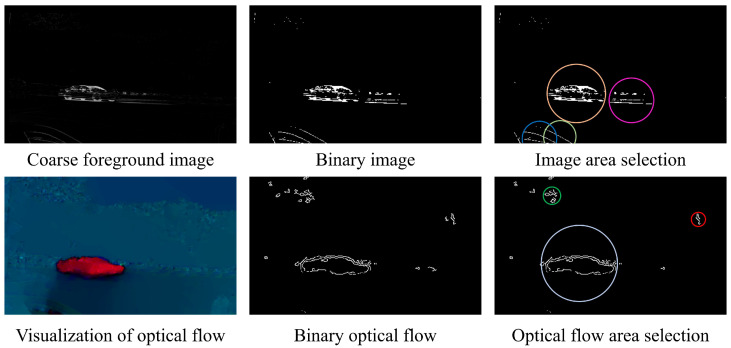
Spatial classification results. Different segmentation areas are indicated by different colours.

**Figure 6 sensors-24-02859-f006:**
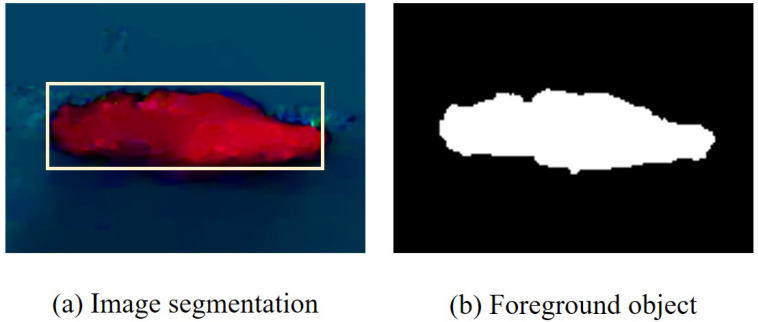
Interactive segmentation result: (**a**) shows the location situation after area selection, and (**b**) applies the GraphCut algorithm to the selected regions.

**Figure 7 sensors-24-02859-f007:**
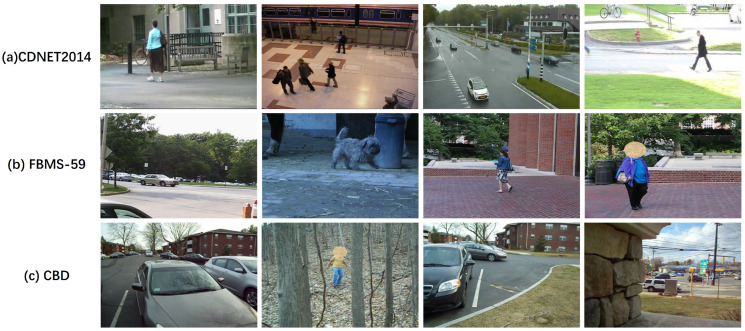
Typical frames of selected datasets.

**Figure 8 sensors-24-02859-f008:**
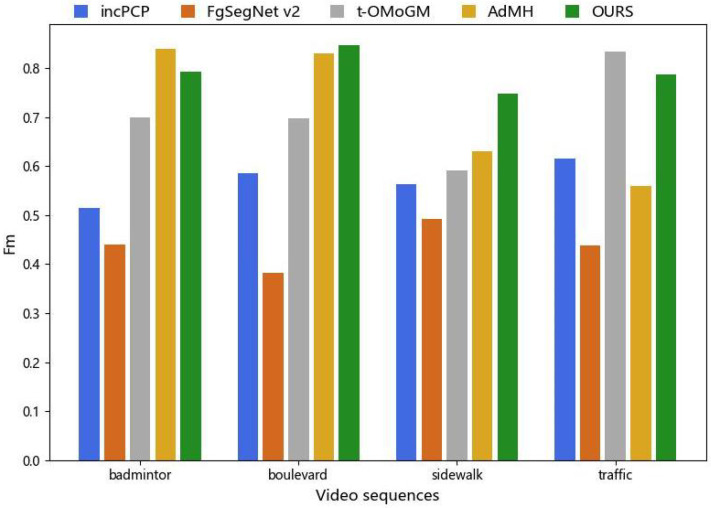
Performance of our method compared with other methods on jittery video sequences.

**Figure 9 sensors-24-02859-f009:**
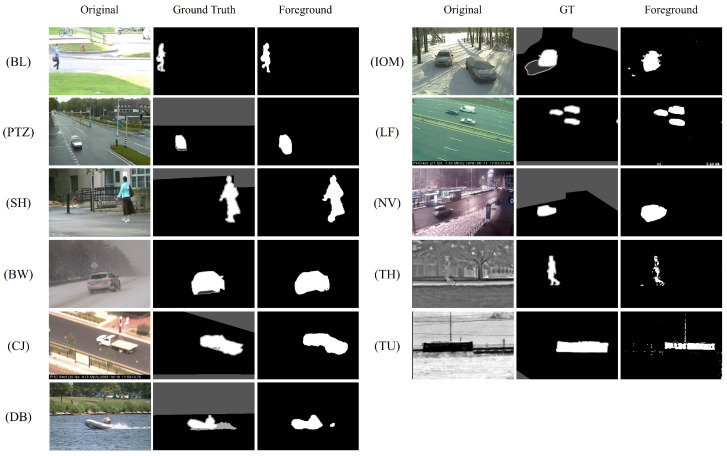
The foreground detection results of our method on the CDNET2014 dataset.

**Figure 10 sensors-24-02859-f010:**
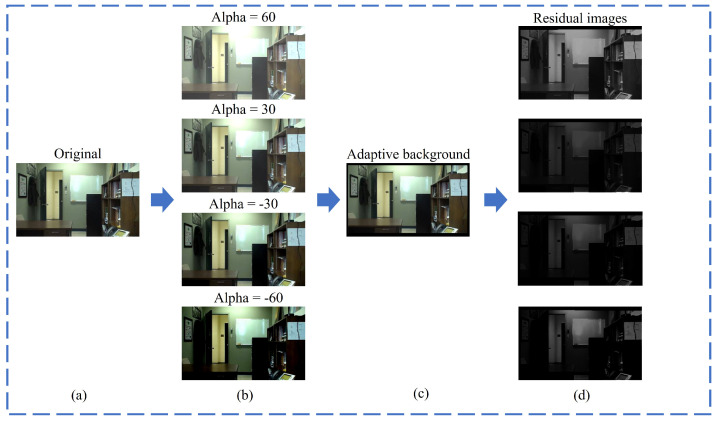
Performance tests with different illumination conditions: (**a**) is the original image sequence. (**b**) is the collection of images processed with different illumination scaling factors (Alpha). (**c**) is the background image obtained by our method under different illumination. (**d**) is the corresponding residual images.

**Figure 11 sensors-24-02859-f011:**
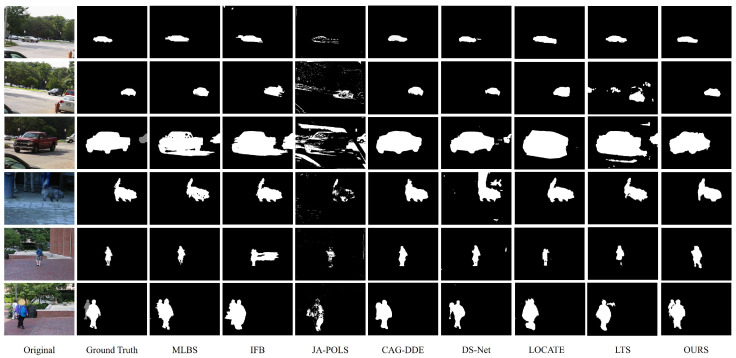
Qualitative comparison of the proposed method with other MOD methods on FBMS-59 dataset. From left to right: Original, Ground Truth, MLBS [57], IFB [58], JA-POLS [37], Sugimura [31], CAG-DDE [51], DS-Net [48], LOCATE [45], LTS [52], and our method.

**Figure 12 sensors-24-02859-f012:**
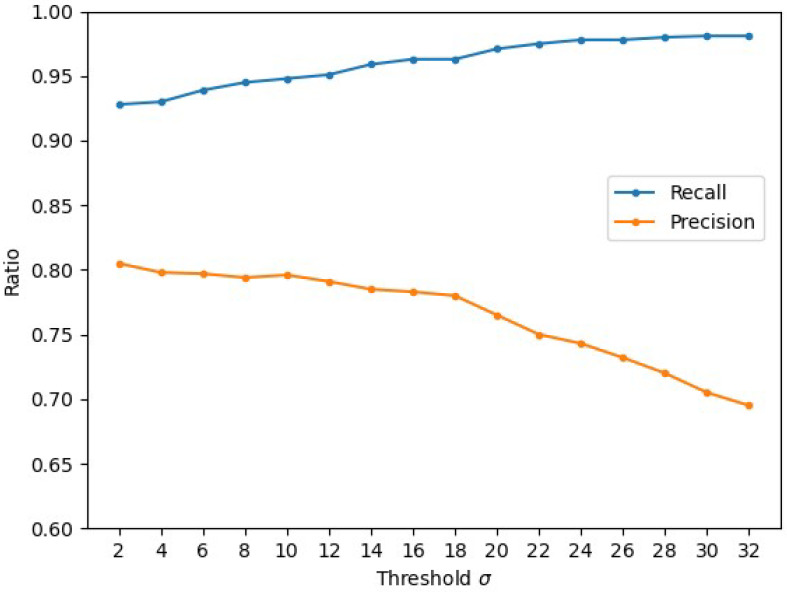
Precision and recall curves for different values of the threshold parameter σ.

**Figure 13 sensors-24-02859-f013:**
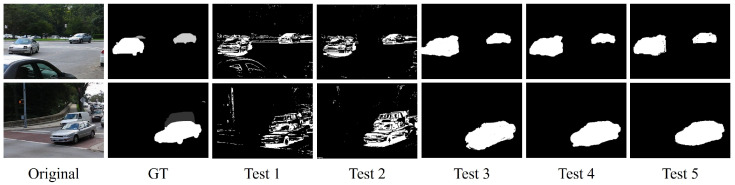
Test 1 represents the usage of only the projection transformation and local background subtraction. Test 2 involves the addition of the motion compensation optimization framework and local background subtraction. Tests 3, 4, and 5 denote the incorporation of local spatial information fusion based on Test 2, employing different interactive graph cut algorithms (GraphCut, OneCut, and GrabCut), respectively.

**Figure 14 sensors-24-02859-f014:**
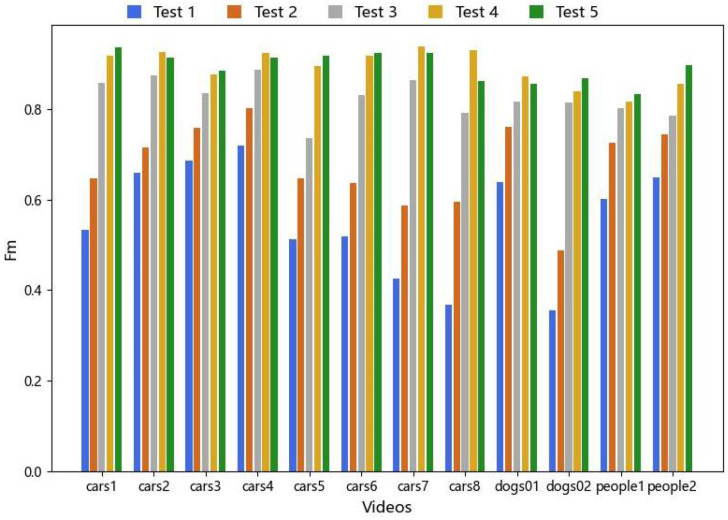
Comparison of qualitative results of ablation experiments.

**Table 1 sensors-24-02859-t001:** Quantitative analysis of the Fm values for each method on the CDNET2014 dataset. To denote the best Fm for each category of videos, use bold formatting, and for the second-best, use underline.

Methods	Videos											
	**BW**	**BL**	**CJ**	**DB**	**IOM**	**LF**	**NV**	**PTZ**	**SH**	**TH**	**TU**	**Overall**
incPCP [7]	0.7324	0.8287	0.5684	0.6844	0.5691	0.5767	0.4736	0.6514	0.7154	0.7436	0.6247	0.6524
SWCD [9]	0.8233	0.9214	0.7411	0.8645	0.7092	0.7374	0.5807	0.4545	0.8779	**0.8581**	0.7735	0.7583
AdMH [10]	0.5600	0.7900	0.7150	0.7750	0.4900	0.4500	0.2600	0.0900	0.6900	0.6900	0.6325	0.5584
FBS-ABL [6]	0.8106	0.8649	0.5298	0.7424	0.7232	0.6328	0.5272	0.3267	0.8671	0.6619	0.5564	0.6585
t-OMoGM [5]	0.7649	0.8027	0.7060	0.7126	0.7348	0.7418	0.5413	0.5843	0.6143	0.7346	0.5466	0.6789
FgSegNet v2 [18]	0.3277	0.6926	0.4266	0.3634	0.2002	0.2482	0.2800	0.3503	0.5295	0.6038	0.0643	0.3715
BSUV-Net [17]	0.8713	**0.9693**	0.7743	0.7967	**0.7499**	0.6797	**0.6987**	0.6282	0.9233	**0.8581**	0.7051	0.7868
OURS	**0.8873**	0.8724	**0.7962**	**0.8759**	0.6751	**0.7618**	0.6584	**0.8032**	**0.9376**	0.7791	**0.7825**	**0.8027**

**Table 2 sensors-24-02859-t002:** Quantitative Analysis of the Fm values for each method on the FBMS-59 dataset. To denote the best Fm for each category of videos, use bold formatting, and for the second-best, use underline.

Methods	Videos												
	**cars1**	**cars2**	**cars3**	**cars4**	**cars5**	**cars6**	**cars7**	**cars8**	**dogs01**	**dogs02**	**people1**	**people2**	**Overall**
MLBS [57]	0.9204	0.9016	0.9316	0.9155	0.8662	0.9223	0.9117	0.8593	0.8200	0.8200	0.8138	**0.9434**	0.8855
IFB [58]	0.6700	0.8167	0.6900	0.8267	0.7333	0.6267	0.7482	0.7103	0.7837	**0.9200**	0.5626	0.7315	0.7350
JA-POLS [37]	0.5325	0.6587	0.6851	0.7189	0.5126	0.5178	0.4548	0.3671	0.6376	0.3149	0.6012	0.6493	0.5542
Sugimura [31]	0.9010	0.9020	**0.9550**	0.8740	0.9090	0.8790	0.9150	0.9210	0.8143	0.8285	0.8020	0.8820	0.8819
CAG-DDE [51]	0.9312	0.9186	0.6605	0.9110	0.4358	0.8634	0.8946	**0.9371**	0.8154	0.8146	0.8738	0.8454	0.8251
DS-Net [48]	0.8922	0.7268	0.4965	0.9089	0.9154	0.8639	0.9146	0.8814	0.7548	0.8459	**0.9074**	0.8154	0.8269
LOCATE [45]	0.8742	0.7589	0.8473	0.6146	0.9018	0.8356	0.8842	0.8174	0.7732	0.7958	0.8092	0.8467	0.8132
LTS [52]	0.7598	0.5178	0.6734	0.5489	0.8067	0.7256	0.5874	0.6793	0.7421	0.7972	0.8126	0.8298	0.7067
OURS	**0.9359**	**0.9225**	0.8846	**0.9231**	**0.9168**	**0.9384**	**0.9389**	0.9293	**0.8530**	0.8661	0.8358	0.8965	**0.9034**

**Table 3 sensors-24-02859-t003:** Quantitative analysis of the Fm values for each method on the CBD dataset. To denote the best Fm for each category of videos, use bold formatting, and for the second-best, use underline.

Methods	Videos					
	**Drive**	**Forest**	**Parking**	**Store**	**Traffic**	**Overall**
MLBS [57]	0.6595	0.7220	0.8366	0.8628	0.4819	0.7126
JA-POLS [37]	0.2101	0.1584	0.1456	0.2876	0.3487	0.2301
Sugimura [31]	**0.8880**	0.8300	0.8110	0.7590	0.5580	0.7690
CAG-DDE [51]	0.0765	**0.8507**	0.4123	0.2458	0.5714	0.4313
DS-Net [48]	0.2654	0.8546	0.1769	0.1137	0.0624	0.2946
LOCATE [45]	0.6813	0.7139	0.4661	0.6764	0.5312	0.6138
LTS [52]	0.3488	0.6795	0.2746	0.5163	0.3867	0.4224
OURS	0.7968	0.8159	**0.8566**	**0.8743**	**0.6381**	**0.7963**

**Table 4 sensors-24-02859-t004:** Average time performance of different interactive graph cut algorithms for one-time segmentation. The best performance is in bold and the next best performance is underlined.

**Method**	GraphCut [63]	OneCut [64]	GrabCut [65]	MLBS [57]	JA-POLS [37]	Sugimura [31]	LOCATE [45]
Time (ms)	7101.96	1369.52	2847.61	6856.73	16,913.64	5178.46	**411.29**

## Data Availability

The datasets used in this study are the publicly available CDNET2014 dataset, FBMS-59 dataset, and CBD dataset. They can be downloaded at the following links: (1) CDNET2014 dataset (http://www.changedetection.net/, (accessed on 15 February 2023)), (2) FBMS-59 dataset (https://lmb.informatik.uni-freiburg.de/resources/datasets/, (accessed on 15 February 2023)), (3) CBD dataset (https://vis-www.cs.umass.edu/motionSegmentation/complexBgVideos.html, (accessed on 15 February 2023)).

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
