# Peer review of "Moving Object Detection in Freely Moving Camera via Global Motion Compensation and Local Spatial Information Fusion"

_sensors, 2024, doi:10.3390/s24092859_

Round 1
Reviewer 1 Report
Comments and Suggestions for Authors
Suggest replacing "..strong.." and "...excellent..." with the figures to compare and contrast the performance, in these statements: "Experimental results on CDNET2014, FBMS-59, and CBD datasets demonstrate that our method exhibits strong performance in challenging scenarios, including shadows, abrupt changes in illumination, camera jitter, occlusion, and moving backgrounds. It showcases excellent adaptability to complex motion environments."
Suggest replacing with "...% of improvement..." instead of claiming "...excellent detection accuracy..", some differences are not distinguishable when comparing OURS to them. - "As can be seen from Table 2, our method outperforms the compared methods in most of 4 the video sequences, demonstrating excellent detection accuracy. "
Suggest replacing the "minor" with a range of values that are considered "minor" in this statement "...as minor discrepancies are insufficient to trigger such detections..."
The future works should be elaborated more so the audience may be interested in reading and citing them for the future directions of their research. Currently, the future opportunities that can be explored are not clear i.e. "In the future, we will plan to address problems related to intermittent motion objects and disguised motion objects."
The writers tend to use the terms "very...", "strong...", "excellent", etc., without providing numerical evidence to support the claims.
It is advisable to look into the vocaburary. For instance ": Motion object detection (MOD) with freely moving cameras has been widely interested in 1 computer vision.", is widely interested used meaningfully here?
Reviewer 2 Report
Comments and Suggestions for Authors
The presented article is devoted to the topical topic of detecting moving objects on video. It contains a description of an original approach based on the application of global motion compensation and the integration of local spatial information.
The article provides an overview of the methods used to solve the problem of detecting moving objects on video. The review presents relevant sources. No self-citations were found. The references are correct.
The article is written in clear language. There are good-level illustrations and a table containing the results of the research. Illustrations and tables greatly facilitate the understanding of the article.
The overall assessment of the article is positive. However, in order to improve it, I can recommend making some corrections and additions, in accordance with the following comments:
1. The above formulas contain parameters for which recommendations on the choice of values are not given.
2. The method of selecting the threshold value for obtaining a binary image is not given. It may make sense to apply one of the adaptive threshold calculation methods.
3. The illustrations show fairly clear images as examples. However, it is not said how the method shows itself in conditions of interference associated, for example, with bad weather conditions (rain, snow, fog, etc.).
4. The article discusses three methods of interactive segmentation. However, it is not said why these methods were chosen.
5. The article shows the execution time of the segmentation methods used. However, the characteristics of the equipment used for the experiments are not specified. In addition, it is advisable to provide characteristics for the total time of object detection.
I think that taking into account the comments made, after minor revision, the article can be recommended for publication.
Reviewer 3 Report
Comments and Suggestions for Authors
The paper proposed a more comprehensive method for moving object detection.
1.The paragraph starting from line 39 on page 2 does not clearly describe the difference between model and motion based methods.
2. The 2nd section "Related Works" does not contain enough state-of-the-art work.
3. The title of Figure 1 does not explain well the figure itsel.
4. The formula (4) has not been clearly explained;
5. The optimization of formula (5) is doubtful. The combintation of Frobenius norm and L2 norm to get a optimized rotation and translation setting need further verification and evidence. This makes the demo results in Figure 3 not evident enough.
6. The sentence "yields the highest number of identical area blocks" at line 293 does not explain the process clearly.
Further, in experiment part the comparisons were made with relatively old methods. Real state-of-the-art methods are needed to be compared to.
Comments on the Quality of English LanguageThe English usage needs to be thoroughly checked.
Round 2
Reviewer 3 Report
Comments and Suggestions for Authors
All previous issues have been addressed. If more comparisons can be done it would better. And, are all parameter settings fixed for all experiments? How about the complexity of the total processing chain?
